# The Chilean Diet: Is It Sustainable?

**DOI:** 10.3390/nu14153103

**Published:** 2022-07-28

**Authors:** Teresita Gormaz, Sandra Cortés, Ornella Tiboni-Oschilewski, Gerardo Weisstaub

**Affiliations:** 1Public Health Nutrition, Institute of Nutrition and Food Technology (INTA), University of Chile, Santiago 7830490, Chile; teresita.gormaz@inta.uchile.cl (T.G.); otiboni@uc.cl (O.T.-O.); 2Departamento de Salud Pública, Escuela de Medicina, Advanced Center for Chronic Diseases (ACCDIS), Pontificia Universidad Católica, Santiago 8330077, Chile; scortesn@uc.cl

**Keywords:** water footprint, carbon footprint, sustainable diets, dietary sustainability assessment

## Abstract

Food systems are one of the main contributors to climate change. Sustainable diets are one strategy to mitigate climate change. Assessments and estimations at a national level are lacking, especially in the Global South, probably due to a lack of national surveys of food consumption and a limited interest in sustainable diets information. The objective of this study is to estimate and describe the carbon and water footprint of the Chilean population’s diet in an overall estimation desegregated by region, age, sex, socioeconomic level and their main characterizations. This study is based on a secondary data analysis from the National Survey of Food Consumption made in 2010. The carbon and water footprint of the food subgroups/person/day were estimated. The results are compared by sex, age group, socioeconomic level, and macro zone. A carbon footprint of 4.67 kg CO_2eq_ and a water footprint of 4177 L, both per person/day, were obtained. Animal-sourced foods, such as dairy and red meat, were responsible for 60.5% of the total carbon footprint and 52.6% of the water footprint. The highest values for both footprints were found in the following groups: men, adolescents, young adults, people with a higher socioeconomic level, and residents in the southern area of the country. The carbon footprint and water footprint values in Chile generated by food consumption would be above the world averages. Transforming the Chilean food system into a more sustainable one with changes in eating patterns is urgently required to attain this transformation.

## 1. Introduction

In the last 50 years, global food production and eating patterns have undergone major changes, moving towards unhealthy, energy-dense, highly processed, and animal-based diets [1]. This change has not only increased the burden of obesity and diet-associated chronic diseases but has also contributed to environmental degradation [2,3]. Food production profoundly affects climate change. Agriculture occupies about 40% of the world’s land, and food production is responsible for up to 30% of global greenhouse gas (GHG) emissions and 70% of freshwater use [4]. The food system defined as “*the elements and activities that are related to the production*, *transformation*, *distribution*, *preparation and consumption of food*” [5] is responsible for 80% of the world’s deforestation, being the main cause of changes in land use and biodiversity loss [6]. Without corrective measures, the environmental cost of the food system could increase between 50–90% in the next three decades [7], making food one of the greatest health and environmental challenges of our time.

In this context, healthy and sustainable eating patterns that generate a lower environmental impact and that contribute to food security and nutrition are required [8]. It is estimated that the transition to more plant-based diets could reduce global mortality (6–10%) and food-related GHG emissions by up to 70% [3]. Diets such as Mediterranean, pescetarian, and vegetarian have reported similar benefits for the population and the environment [2] since they are all based on plants and have weekly limitations on animal-source foods [4]. Different technical organizations (EAT Lancet [4], FAO 2019 sustainable diets [8], and the Barilla Center for Food and Nutrition [9]) have recommended diets based on the above criteria because they contribute to both human and planetary health. Until now, very few countries have included sustainability criteria in their dietary guidelines, such as reducing the consumption of red meat and encouraging the consumption of plant-based and minimally processed foods [10,11].

The climate impact of food production can be measured by different indicators. The number of GHG generated (CO_2eq_) and freshwater consumed (L) throughout the different stages of the production chain are two of the most commonly used forms [9]. International evidence has evaluated the annual GHG emissions per capita of the average diets of some countries, from analyses carried out from national balance sheets, and direct/indirect consumption or food purchases per household, among others. It has been identified that developed countries with a high consumption of meat and processed foods would have the highest water and carbon footprints [12,13] and countries with more traditional and local diets the lowest [14].

In Chile, the food and nutritional situation is complex. A total of 74% of the adult population is overweight [15], which would be associated with a high “ecological cost of obesity”, a concept that refers to the environmental impact attributed to the increased demand for unhealthy foods [16]. In addition to this, our country ranks second in the consumption of ultra processed foods in the region [17] and is among the 10 countries with the highest meat consumption per capita on the planet [18]. The United Nations Framework Convention on Climate Change defines Chile as a country vulnerable to climate change, and the IPCC points out that urgent transformations are required in Chile because by the year 2025 it will be one of the 30 countries with the highest water risk in the world [19]. To date, “*almost 80% of the national territory is affected by drought*” [20]. Therefore, it is essential to know the environmental impact of the Chilean diet for better management of the environmental and nutritional crisis.

The objective of the study was to estimate the carbon and water footprint associated with the diet of the Chilean populations evaluated in 2010. These estimates are carried out globally and disaggregated by different sociodemographic characteristics.

## 2. Materials and Methods

### 2.1. National Survey of Food Consumption

A secondary analysis in public information was carried out by using the statistical descriptor generated by the 2010 National Food Consumption Survey of Chile (ENCA) (Minsal, 2010). The ENCA was a population survey of 4920 people older than 2 years old, with national representativeness including both urban and rural areas. The consumption results for the whole Chilean population were obtained through a quantified consumption trend survey considering the type, frequency, and amount of each food eaten during the month before the day of the interview. From the final report of the survey, the values found on the percentiles 25, 50, and 75 of the consumption of the food subgroups used in the ENCA (*n* = 30) were collected according to the general population and to sex, age group, socioeconomic level, and macrozones (Appendix A). Individual consumption values were not considered. The unit of analysis was the consumption of g/day or ml/day of the different food subgroups, from which the carbon and water footprints were estimated.

### 2.2. Carbon Footprint and Water Footprint

The carbon footprint (CF) was defined as the number of GHG generated (CO_2eq_) in the different processes of the food production chain. The water footprint (WF) was defined as the amount of freshwater consumed throughout the different stages of the production chain (planting, harvesting, transportation, packaging, distribution, etc.) [9]. Both indicators are quantified using the “life cycle analysis” (LCA) methodology [21,22]; all the data used refers to an analysis from the supply of raw materials to the main distribution point. This information was not available for the Chilean context.

### 2.3. Bibliographic Search for the Calculation of the CF and WF

To carry out the aforementioned bibliographic search, the publications made since 2000 on the Pubmed, ScienceDirect, Lilacs, Trip database, Scielo, Google Scholar, and Epistemonikos sites were considered, defining as the search strategy: (“carbon footprint” OR “water footprint “ OR “environmental footprint”) AND (“diet” OR “food” OR “dietary intake”). In addition, the names of the different subgroups and foods reported in the ENCA were used to search for specific footprints that were not found. The studies selected for this research provided further articles by the references used. The gray literature obtained in the form of industry reports, projects, and government documents was also obtained in this way when the method before did not provide the needed results.

The selected studies were those that had an LCA methodology [23] and that declared the carbon and water footprint generated from the supply of raw materials of production to the finished product at the main wholesale distribution point. A total of 11 studies were selected to build the database used for this study.

For the carbon footprint, 4 bibliographic sources were selected: (1) a systematic review and meta-analysis [24] for 168 types of fresh foods; (2) a database for processed products [25]; (3) a Chilean database from Fundación Chile, and the Ministry of the Environment [26] with values for 16 foods from the agri-food and wine sector export, such as apple, table grape, blueberry, avocado, plum, mussel, salmon, chicken, pork, wine, gouda cheese, milk powder, frozen raspberry, dehydrated apple, and canned apple and peach juice; and (4) the gray literature from the noncaloric sweeteners industry [27]. For the water footprint, 6 bibliographic sources were selected: (1) a global database for farmed foods [28]; (2) animal products [29,30]; (3) sugar-sweetened beverages [31]; (4) seafood [32]; (5) soy products and derivatives [33]; (6) and the gray literature from industry documentation for noncaloric sweeteners [27].

The total foods found (*n* = 197) were ordered by the same subgroups provided by the ENCA (*n* = 30) in a database created for this study. For each food, the average footprint value was calculated from all the databases containing that food’s footprint. For those foods that did not have information in the selected database, their value was omitted, and the average of their subgroup was considered. For example, within the “red meat” subgroup, if there was no data for “*pork ossobuco*”, the average of the red meat subgroup was used. The ENCA values used for this study are the 50th percentile (Appendix B)

From this bibliographic search, the values per 100 g/mL of food were collected and the average value of the carbon and water footprint was calculated for each of the 30 food subgroups of the ENCA (Appendix C).

### 2.4. Analysis of the Results

A descriptive analysis of the results was performed. First, the CF and WF were calculated for the amounts consumed per day of each food subgroup reported in the ENCA (CF and WF for those g or mL/day) for the general population. Later, both footprints were calculated by sex, socioeconomic level, age group, and macrozone. This provided the total values/day and the contribuiting percentage of each food subgroup to the total CF and WF.

## 3. Results

Food consumption in Chile is estimated to account for 4.67 kg of CO_2eq_ per person daily and uses 4177 L of water per person daily.

### 3.1. Carbon Footprint

A total of 60.5% of the CF is generated by animal source foods, mainly cheeses, and red and processed meats (Figure 1). Furthermore, given its high consumption, bread is responsible for one-sixth of the CF. Men show a 28.5% higher carbon footprint than women (4.4 v/s 3.2 Kg CO_2eq_ per person/day, respectively) and a higher net consumption of processed meat, bread, and red meat. When analyzing the CF by age group, we observe that the values of young adults are 33% higher than those found in preschoolers and older adults, at the same time that young adults and school children have the highest consumption of cheese and red meat, respectively. The highest socioeconomic level presents a CF 16% higher than the lowest socioeconomic level (Figure 2). When comparing the CF by macrozone, the southern zone had higher values (13% more) than the northern.

### 3.2. Water Footprint

In the case of the water footprint, 52.6% comes from animal source foods, and the main contributors were dairy products (high and low-fat), red and processed meat, cheese, and candy and sweet foods (Figure 1). Men had a 24% higher water footprint than women (866 L per person/day more). When analyzing WF by age group, secondary school students are the group with the highest demand for water resources, being 28.6% higher than the group of older adults, which registers the lowest figures.

In reference to the socioeconomic level, the higher the socioeconomic level, the greater the contributing percentage of animal source foods in the diet, and therefore, the greater the water footprint (16% higher WF than the lowest socioeconomic level) (Figure 2). When comparing WF by macrozone, slightly higher values are observed in the southern zone of the country.

Finally, the results of Chile obtained in this study were compared with the “planetary diet” proposed by the EAT–Lancet Commission (EAT, 2019). From this comparison, it was observed that there are groups of foods that greatly exceed the indicated recommendations. Red meat exceeds the recommendation by 443%, sugars by 243%, and dairy products by 235%, in contrast to other food groups where their consumption is much lower than recommended, such as dried legumes and soy with 23%, oils and fats at 37.8%, and nuts at 38.2% (Figure 3).

## 4. Discussion

The Chilean diet produces 4.67 kg of CO_2eq_ per person daily, a value 42% higher than what is estimated would generate a nutritionally adequate and ecologically sustainable diet worldwide [34]. When comparing the Chilean CF with that of other countries, it is observed that it is lower than that of the USA (4.72 Kg CO_2eq_) [12] and Argentina (5.48 Kg CO_2eq_) [35] but higher than that of Peru (2.61 Kg CO_2eq_) [14], France (4.09 Kg CO_2eq_) [36], Brazil (4.48 Kg CO_2eq_) [37], and Denmark (4.63 Kg CO_2eq_) [13], differences produced probably by the high consumption of animal-sourced foods. In addition, the CO_2_ emissions generated by the Chilean diet represent 33% of the country’s total GHG emissions [38], which would be above the estimated average value of CF associated with food systems (20–30%) [4].

On the other hand, the WF generated by food consumption in Chile requires a daily use of 4177 L of water per person, a value 23% higher than the estimated world average [39]. To have an idea of the magnitude of the value mentioned, the amount of water consumed according to our estimates, which is required to produce the amount of food that a person consumes daily, is the equivalent of that required to bathe for 41 days for 5 min a day. Compared with other countries, the WF of the Chilean diet would be higher than the Nordic countries (3233 L of water/capita/day: Lcd) [40] and Brazil (3507 Lcd) [41], similar to the average occupied by the European Union (4265 Lcd) [42] and less than the USA (6795 Lcd) [39].

The high values for carbon and water footprints observed in men have also been described in several countries [34]. In the present study, the difference is mainly associated with the greater consumption of red and processed meats by men compared with women (36.1% more CF in processed meats and 30.9% more in red meats) and the greater amounts of food consumption in general (34.9% more consumption of bread, 27.8% more cereals, and 18% more dairy products than women).

Regarding the differences by age group, it was observed that young adults were those with the highest dietary CF and WF, similar to what occurs in other countries such as Brazil [37] and Sweden [34]. In our results, the highest socioeconomic level is the one that reports the highest CF and WF, which could be aligned to other national level studies where higher incomes have a greater environmental footprint due to their diet [12,13]. This is especially worrying considering that the economy is expected to continue to grow, and groups with lower socioeconomic levels tend to have consumption patterns more similar to those of higher socioeconomic levels, which tend to have an even greater environmental impact at a general level.

According to the macrozone, it can be seen that the largest carbon and water footprints are found in the southern zone. Although it is beyond the scope of the study to analyze the impact of specific foods, it should be noted that this difference is attributed to the fact that the southern zone is the one with the lowest consumption of fruit and vegetable groups and the highest consumption of cheeses, red meats, and processed and saturated fats from the macrozones.

Finally, it is important to comment that according to the comparisons made with the “planetary diet” of the EAT–Lancet Commission, Chile would have a diet that is inconsistent with the environment and human health.

In this approach, at the level of Chile as a country, and given the limitations of the available data, the nutritional characterization was out of scope. However, from this survey, it is stated by the authors that 95% of the population does not comply with the dietary recommendations. This situation shows the inequalities in the Chilean population, as it is worse in the rural population and lower socioeconomic level. Undoubtedly, the availability of updated information on nutritional quality, and also on other factors such as the chemical and toxicological composition of the diet used in Chile, represents a limitation of the results presented. Similarly, it is necessary to improve the collection of systematic information regarding the costs of the diet described for the Chilean population. It is recognized that those foods of animal origin, such as red meat, have a higher economic value per kilogram and are also more susceptible to national and international market changes. This scenario can be anticipated given the higher cost of healthy diets in the Latin American and Caribbean region compared with other regions: USD 3.89 per person per day in America, Asia (USD 3.72), Africa (USD 3.46), North America and Europe (USD 3.19), and Oceania (USD 3.07). In the current scenario of the occurrence of other environmental social crises, it is estimated that at least 22.5% of the region’s population, estimated at 131.3 million people, cannot access a healthy diet [43].

It is important to note that the results provided come from the ENCA 2010, when the national overweight and obesity prevalence was 64%, lower than today’s 74% [15]. In addition, the data is not adjusted for possible underestimation of any consumption survey, therefore the estimated environmental impact could be even greater. The CF and WF were estimated based on the data available in the literature and were not specifically associated with the food system in Chile, using various secondary sources to obtain these traces due to the lack of local evidence for all foods. Some specific foods (typical of the Chilean culture or others uncommon globally, such as pisco liquor, pork rinds, hard candy, murtilla, turtledove beans, tongue beef, donkey milk, etc.) were not found in the literature, so in these cases their value was omitted and the value of the subgroup was calculated, which could alter the results. However, their consumption is very low, thus probably the results would not change too much. Another aspect to consider is that the use of added data does not allow conclusions to be applied at the individual level, and the results are not adjusted for calorie intake. Among the strengths, it is worth noting that this is the first work on the environmental impact of the Chilean diet, with national representativeness. In addition, the data used on consumption were retrieved from a nationally representative survey. The data is segregated at different levels such as socioeconomic levels, age group, sex, and macrozone and includes several food groups, even more than those typically mentioned in the national dietary guidelines, which allows a better environmental estimation. However, it is important to consider that results are from a secondary analysis from the ENCA which is from 2010, so results can be different and theoretically higher nowadays.

Possible solutions for improvement to better estimate the carbon footprint or the water footprint could incorporate data from other national representative studies, as has been evaluated in contexts relatively similar to Chile. An example of this is the work of [44], where an individual-type approach was used, in which the daily intake per person was valued, establishing differences with an indicator of healthy eating associated with a reduction in mortality. In this proposal, in addition to the estimation of the carbon footprint, other indicators of environmental sustainability were also established. Opening the debate in a more comprehensive way would allow setting more ambitious goals for the Latin American countries, in pursuit of the sustainability of food production for a healthy diet [45].

Positioning a healthier eating pattern consistent with the environment is the next great challenge we have as a country. For this, red meat and dairy consumption should be reduced, while increasing the consumption of legumes, whole grains, vegetables, seeds, and other alternatives of vegetable origin. Promoting a diet based on the abundant resources of the earth that we have as a country is one of the most coherent options.

This issue should be included in health promotion strategies, in healthy lifestyle guides, in food guides, and in the creation of public policies such as taxes on the ecological cost of food or eco-labeling.

## 5. Conclusions

Chile is in a conflictive scenario both for the types of food most consumed by the population and for the amount consumed per person.

It should be considered that the high intake of ultraprocessed foods and red meat, characteristic of several countries in the region, increases GHG emissions and is also detrimental to human health. Economic policies that impact the sale and production of unhealthy and healthy foods, as well as those that modify the food environment, could be an essential tool to reduce the consumption of ultraprocessed foods.

If we analyze the aspects mentioned in the Sustainable Development Goals (SDG) proposed by the United Nations and accepted by the Member States, we urgently need to rethink our food system. Food sustainability and agri-food systems should be included in the public policies and strategies of different ministries, such as health, agriculture, environment, economy, fishing, social development, science and technology, gender equity, and education. These policies must not only consider the availability and access to healthy food but also their environmental sustainability, especially in a political and social context where the Right to Adequate Food and Food Sovereignty is the subject of constitutional discussion.

Chilean nutritional sustainable recommendations are urgent to be considered. Understanding that human health is not isolated from planetary health is essential for the survival of different species and for the quality of future human life.

## Figures and Tables

**Figure 1 nutrients-14-03103-f001:**
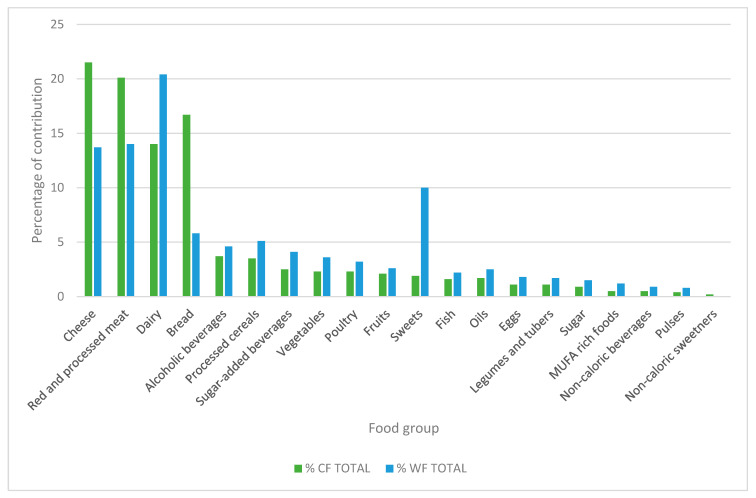
Percentage of contribution of food subgroups to the total carbon and water footprint of the Chilean diet for the general population. The main contributors are cheese (21.5% CF, 13.7% WF), red and processed meat (20.1% CF, 14% WF), dairy (14% CF, 20.4% WF), bread (16.7% CF, 5.8% WF), and “sweets” (originally in the survey as sugar from sweets and other sweet food) (1.9 CF%, 10% WF).

**Figure 2 nutrients-14-03103-f002:**
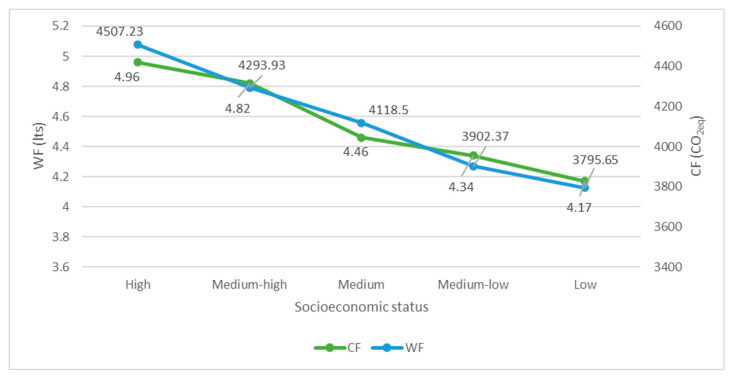
Carbon footprint (Kg CO_2_eq/day/person) and water footprint (L/day/person) of diet according to socioeconomic status.

**Figure 3 nutrients-14-03103-f003:**
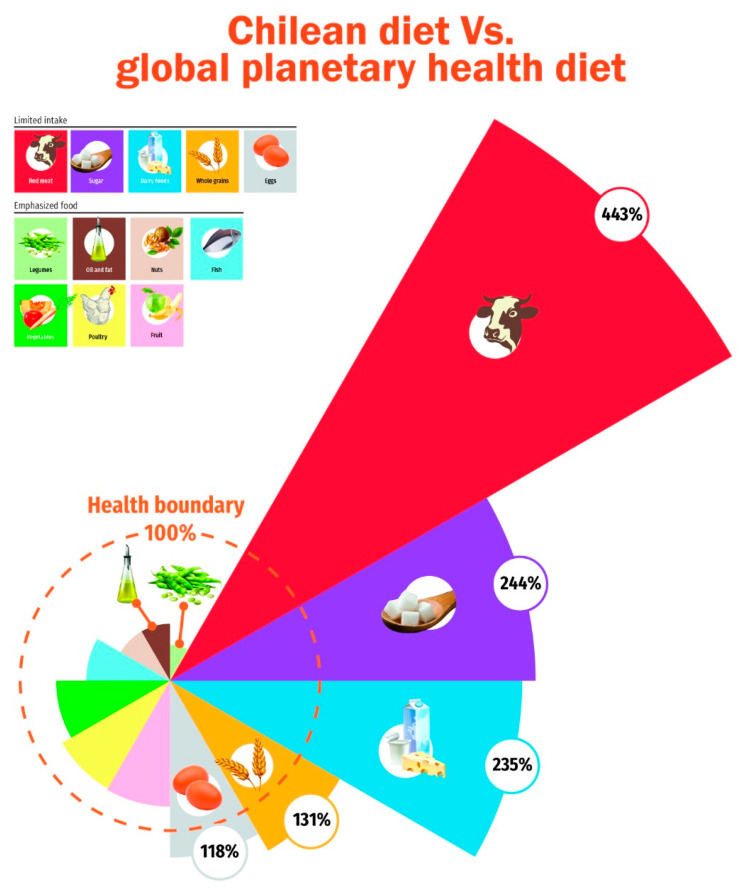
The “diet gap” between current Chilean dietary patterns and the Planetary health diet defined by EAT–Lancet Commission [4]. The Health boundary is a target defined as the safe operating space for food systems, human health, and environmental sustainability. Food groups outside this health boundary are unsustainable. Graphic credit to http://www.eatforum.org (accessed on 4 July 2022).

## Data Availability

https://www.minsal.cl/encadescarga/, accessed on 4 July 2022.

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
