# Peer review of "The Chilean Diet: Is It Sustainable?"

_nutrients, 2022, doi:10.3390/nu14153103_

Round 1

Reviewer 1 Report

We appreciate sending this article since there is not much information published about South American countries in general and about Chile in particular on this subject.

In general, the article is well structured, although for its publication some contributions of results in the article are recommended, improve the information handled in some aspects and provide supplementary data. Overall, all of this would reasonably improve the quality of the results reported, as well as their transparency, an issue that is also desirable. These demands are detailed below:

1. Introduction

The introduction presents the object of study reasonably well and justifies the purpose of the study. Certainly there is little information on South American countries on this subject

2. Objective: It would be more precise to specify that the estimates will be carried out globally and disaggregated by different sociodemographic characteristics of the Chilean population.

3. Consumption data (25th, 50th and 75th percentiles) are not provided in the supplementary material. This would be the same and even more interesting than the mere description of the sociodemographic variables used (Appendix A, Supplementary material 1)

4. Likewise, supplementary material 2 shows the environmental footprint values ​​used and the references that support them. Perhaps it would be equally important to provide tables with the 75 percentiles of consumption of the food groups on which these values ​​were used as supplementary material.

5. Throughout the text there are some references in APA style. For example “(BCFN, 2016)” on line 96 and “(Global Pact, 2015)” on line 116 of the material and methods section, which have also not been included in the list of references

6. Some reflection or mention of the foods whose consumption values ​​were assigned the mean of the group would be interesting, since, if the absence of these values ​​is due to their infrequent consumption, this would mean an overestimation of the footprint environment of that group.

7. Figures or tables of the results disaggregated by sex, age or macro area are not provided, only the most significant results are referred to in the text. However, it is provided for socioeconomic level.

8. Discussion. It would be interesting to compare their results with countries with cultural and socioeconomic proximity since these factors influence the pattern of consumption (only Peru and Brazil appear). Items are recommended:

Arrieta, E. M., Geri, M., Coquet, J. B., Scavuzzo, C. M., Zapata, M. E., & González, A. D. (2021). Quality and environmental footprints of diets by socio-economic status in Argentina. The Science of the total environment, 801, 149686. https://doi.org/10.1016/j.scitotenv.2021.149686

Castellanos-Gutiérrez A, Sánchez-Pimienta TG, Batis C, Willett W, Rivera JA. Toward a healthy and sustainable diet in Mexico: where are we and how can we move forward? Am J Clin Nutr. 2021 May 8;113(5):1177-1184. doi: 10.1093/ajcn/nqaa411.

9. References: The references used are valid and consistent with the topic. However, the checklist requires major revision as there is diversity in the format used. Some references lack authors. Can you communicate what style of references you have used? Nutrients recommends ACS Style https://pubs.acs.org/doi/full/10.1021/acsguide.40303

Author Response

Thank you very much for your excellent comments. Attached is the new manuscript with track changes. There you can see the improvements thanks to your comments, where we thought could be better within the manuscript. Thanks again for your valuable comments.

Reviewer 2 Report

The authors have done a fine job developing and presenting their data. With some additional work the paper could have been even more valuable.  More input on solutions that are feasible within the population (i.e., cost of changes, accessibility to foods, respecting food culture) would be appreciated. 

One point missing from the evaluation of a 'healthy, sustainable diet' was nutrition.  Examining the diet based solely on environmental footprint overlooks that nutrient dense foods often have higher environmental impacts (e.g., higher protein quality associated with animal products including egg and dairy).

What was the population's nutrient intake level associated with the different diets?  Are these data available for the population in Chile?

Discussion of socioeconomic level and diets seemed incomplete without comments on the cost of current vs proposed ideal diet.  

Author Response

Thank you for your excellent comments. The manuscript has been updated with track changes, which is attached. Also, your comments have been answered within the text. Thanks for your comments again.

Reviewer 3 Report

The manuscript investigates the sustainability of Chilean eating habits estimating the carbon and water footprint. The article is very interesting and well structured; however only a modification is required:

 1)      Insert the Appendix B, Supplementary material 2 into the text (in materials and methods).

Author Response

In track changes the comment resolved. Thank you
